# Shuterin Enhances the Cytotoxicity of the Natural Killer Leukemia Cell Line KHYG-1 by Increasing the Expression Levels of Granzyme B and IFN-γ through the MAPK and Ras/Raf Signaling Pathways

**DOI:** 10.3390/ijms232112816

**Published:** 2022-10-24

**Authors:** Jen-Tsun Lin, Yi-Ching Chuang, Mu-Kuan Chen, Yu-Sheng Lo, Chia-Chieh Lin, Hsin-Yu Ho, Yen-Tze Liu, Ming-Ju Hsieh

**Affiliations:** 1Department of Medicine, Division of Hematology and Oncology, Changhua Christian Hospital, Changhua 500, Taiwan; 2Department of Post-Baccalaureate Medicine, College of Medicine, National Chung Hsing University, Taichung 402, Taiwan; 3School of Medicine, Chung Shan Medical University, Taichung 40201, Taiwan; 4Oral Cancer Research Center, Changhua Christian Hospital, Changhua 500, Taiwan; 5Department of Otorhinolaryngology, Head and Neck Surgery, Changhua Christian Hospital, Changhua 500, Taiwan; 6Department of Family Medicine, Changhua Christian Hospital, Changhua 500, Taiwan; 7College of Medicine, National Chung Hsing University, Taichung 402, Taiwan; 8Graduate Institute of Biomedical Sciences, China Medical University, Taichung 404, Taiwan

**Keywords:** shuterin, natural killer cells, MAPK pathway, HNSCC, INF-γ

## Abstract

Natural killer (NK) cell therapy is an emerging tool for cancer immunotherapy. NK cells are isolated from peripheral blood, and their number and activity are limited. Therefore, primary NK cells should be expanded substantially, and their proliferation and cytotoxicity must be enhanced. Shuterin is a phytochemical isolated from *Ficus thonningii*. In this study, we explored the possible capacity of shuterin to enhance the proliferation and activity of KHYG-1 cells (an NK leukemia cell line). Shuterin enhanced the proliferation of KHYG-1 cells and their cytotoxicity to K562 cells. Moreover, this phytochemical induced the expression of granzyme B by promoting the phosphorylated cyclic adenosine monophosphate response element–binding protein (CREB) and mitogen-activated protein kinase (MAPK) signaling pathways. Furthermore, the secretion of interferon (IFN)-γ increased with increasing levels of shuterin in KHYG-1 cells and NK cells obtained from adults with head and neck squamous cell carcinoma. Shuterin appeared to induce IFN-γ secretion by increasing the expression of lectin-like transcript 1 and the phosphorylation of proteins involved in the Ras/Raf pathway. Thus, shuterin represents a promising agent for promoting the proliferation and cytotoxicity of NK cells.

## 1. Introduction

The primary modalities for tumor treatment include surgery, chemotherapy, and radiotherapy; however, drug resistance often reduces the survival rate of patients with cancer [1]. Immunotherapy involves targeted tumor treatment through the activation of specific immune responses. In autologous immune enhancement therapy, autologous immune cells are cultured in vitro and then injected back into the body to eradicate cancer cells. Immune cells, such as natural killer (NK) cells (e.g., γδ and αβ T cells), cytotoxic T lymphocytes (CTLs), and dendritic cells (DCs), have various clinical applications. Clinical use of Autologous Immune Enhancement Therapy (AIET) to fight against cancer. In addition to injecting expanded activated NK cells, which can combine the characteristics of different immune cells to improve the advantages of immunotherapy, CTLs target cancer cells expressing a major histocompatibility complex (MHC), and NK cells also fight against cells that do not express MHC, such that the antigen-stimulated peptide-pulsed DCs can activate CTLs. The combination of the innate and adaptive immune systems could be a tool for cancer immunotherapy [2].The effective enhancement of the number and activity of NK cells and CTLs is closely associated with satisfactory prognosis, reduced tumor size, and prolonged survival in cancer [3,4].

NK cells play an irreplaceable role in the immune network: immune surveillance for internal pathogens and the prevention of infections caused by external pathogens [5]. The primary mechanism underlying the cytolytic activity of NK cells involves the release of perforin and granzyme (e.g., granzyme A, B, and C); other mechanisms include pathways involving Fas ligand (FasL) and tumor necrosis factor (TNF)–related apoptosis-inducing ligand (TRAIL) [6,7,8,9]. Granzyme B is a key enzyme in NK cells that helps trigger various cytotoxic effectors. After endocytosis by target cells, this enzyme helps activate the caspase pathway and regulate mitochondrial functions, resulting in nuclear condensation and proteolysis, which induce the fragmentation of target cell DNA, ultimately leading to apoptosis [10]. The expression of granzyme B is regulated by the following transcription factors: activator protein 1, cyclic adenosine monophosphate response element–binding protein (CREB), Ikaros, and C-repeat binding factor [11,12]. Furthermore, NK cells secrete cytotoxic cytokines, particularly interferon γ (IFN-γ), which acts directly on the target cells [13].

Natural phytochemicals may serve as a source of new drugs for enhancing the antitumor response of immune cells. For instance, a plant lectin—viscum album agglutinin I—enhances the proliferation and cytotoxicity of NK cells [14]. Curcumin enhanced the cytotoxicity of NK92 cells against K562 cells and increased the surface expression of CD16 and CD56 in NK92 cells [15]. Resveratrol is a polyphenol phytoalexin of plants; research indicated that in NK92 cells treated with resveratrol increased the toxicity of the target cell and that in human NK cells resveratrol had a stimulatory effect of INF-γ secretion [16,17]. Shuterin was first defined as the diffuse production of fungal inoculation leaflets from *Shuterza uestita*. Shuterin is one of 21 reported phytochemicals isolated from the fruits and roots of *Ficus thonningii*. The plant extract of *F. thonningii* exhibits anti-inflammatory, antibacterial, and antidiabetic activities [18]. Shuterin is also a phytoalexin that belongs to the group of 3-hydroxyflavanone (flavonol) phytoalexin, which strongly inhibits the growth of *Cladosporium herbarum* [19]. However, the possible immune-boosting capacity of shuterin remains obscure.

KHYG-1 is an NK leukemia cell line that is cytotoxic to K562 cells and was first established by Yagita et al. [20,21]. We used the KHYG-1 cell line to study the regulatory pathways involved in the enhancement of NK cell cytotoxicity. Our knowledge regarding the effects of shuterin on KHYG-1 cells remains limited. Therefore, this study centered on exploring the mechanisms underlying the effects of shuterin on the proliferation and activity of KHYG-1 cells.

## 2. Results

### 2.1. Effects of Shuterin on the Proliferation and Cytotoxicity of KHYG-1 Cells

Figure 1A shows the chemical structure of shuterin. KHYG-1 cells were treated with various doses (0, 1, 5, and 10 µM) of shuterin for 72 h and subjected to WST-8 assays (Figure 1B). Shuterin substantially increased the proliferation of KHYG-1 cells. Shuterin markedly increased the activity of KHYG-1 cells and thus increased these cells’ cytotoxicity to K562 cells. At an E:T ratio of 6:1, cytotoxicity increased from 40% to 54% within 2 h (Figure 1C).

### 2.2. Effects of Shuterin on the Expression Levels of Granzyme B and IFN-γ

IFN-γ secreted by NK cells plays a pivotal role in antitumor and antiviral responses [22,23]. The secretion of IFN-γ from KHYG-1 cells was evaluated through an enzyme-linked immunosorbent assay. Shuterin markedly enhanced the secretion of IFN-γ in a dose-dependent manner (Figure 1D).

Next, the mRNA levels of cytotoxic effectors were examined by real-time PCR analysis. The effects of shuterin on the transcriptional activation of perforin, granzyme A, granzyme B, FasL, granulysin, and IFN-γ were examined. The results showed that the target genes were significantly enhanced by shuterin, except for perforin. The expression of granzyme A and granzyme B increased markedly (2.9-fold) and (3.8-fold), and the other mRNA levels modestly increased (Appendix A). Furthermore, protein levels were analyzed by Western blotting in KHYG-1 cells treated with shuterin for 24 h and those left untreated. At a concentration of 10 μM, shuterin considerably increased granzyme A, granzyme B, FasL, and granulysin by 240%, 350%, 128%, and 187%, respectively, but not perforin (Figure 1E,F).

### 2.3. Effects of Shuterin on CREB Phosphorylation and Histone Deacetylation

From our data, shuterin significantly increased granzyme B expression at the mRNA and protein levels. Phosphorylated CREB transcription factor regulates the expression of granzyme B [24]. We examined the effect of shuterin on CREB phosphorylation. Shuterin markedly increased the levels of phosphorylated CREB in the KHYG-1 cells (Figure 2A,B). Histone acetylation and deacetylation are crucial for chromatin regulation and gene expression in eukaryotic cells [25]. Shuterin substantially increased the expression of acetyl-histone H3 (Lys9/Lys14) (Figure 2C,D). The chemical reagent trichostatin A (TSA) is a histone deacetylase inhibitor (HDACi) [26]. We pretreated with TSA (100 nM) for 1 h, followed by shuterin addition, which more significantly increased the expression of acetyl histone H3 but did not increase the expression of granzyme B (Figure 2E,F).

### 2.4. Effects of Shuterin on the Activation of Caspase-Mediated Apoptosis

Cytotoxic granules secreted by CTLs and NK cells (e.g., granzyme B) disrupt mitochondrial membrane potential and induce caspase-mediated apoptosis [8,27]. Shuterin exerted no cytotoxic effects on the K562 cells after treatment for 24 h (Figure 3C). However, in the coculture of KHYG-1 with K562 cells, shuterin markedly increased the proportion of depolarized cells in a dose-dependent manner (Figure 3A,B) and the levels of cleaved caspase-3, caspase-9, and granzyme B (Figure 3D,E).

### 2.5. Shuterin Increases Granzyme B Expression via the MAPK Pathway

Shuterin substantially increased the expression of granzyme B. However, the treatment of KHYG-1 cells with Z-AAD-CMK considerably reversed the shuterin-induced increase in the expression of granzyme B (Figure 4A,B).

Previous reports have indicated that the control of human granzyme B gene transcription occurs through the MAPK signaling pathway [28] and NF-κB binding site [29]. We tested the effect of shuterin on the protein expression of NF-κB, and the results showed that shuterin did not change the protein expression of NF-κB. The assessment of PI3K/AKT levels and the MAPK signaling pathway in KHYG-1 cells revealed that shuterin substantially increased the phosphorylation of ERK1/2, JNK, and p38 but not of AKT (Figure 4C,D). The cells were treated with AKT, ERK, JNK, and p38 inhibitors (LY-294002, U0126, SP-600125, and SB-203580). Cotreatment with shuterin and MAPK inhibitors reduced the expression of granzyme B in KHYG-1 cells compared with shuterin alone (Figure 4E–H).

### 2.6. Shuterin Increase IFN-γ Expression Is Associated with a Higher LLT1 Receptor and Ras/Raf Pathway

Lectin-like transcript 1 (LLT1)–mediated activation of IFN-γ production by human NK cells is involved in the MAPK kinase (MEK) pathway [30]. We explored the possible association between shuterin and IFN-γ expression, we evaluated the expression of LLT1 and the phosphorylation of proteins involved in the Ras/Raf signaling pathway in KHYG-1 cells (Figure 5A,B). Shuterin substantially increased the expression levels of LLT1 and Ras and the phosphorylation of Raf and MEK in KHYG-1 cells. As mentioned, shuterin considerably increased the expression of IFN-γ and the cytolytic activity of KHYG-1 cells. NK cells obtained from adult patients with HNSCC were also subjected to similar assessments. Shuterin markedly increased the expression levels of the cytotoxic effectors granzyme A and granzyme B and the phosphorylation of ERK and CREB in NK cells obtained from adult patients with HNSCC (Figure 5C and Appendix A). Furthermore, shuterin considerably increased the expression of IFN-γ is associated with a higher LLT1 receptor and Ras/Raf pathway (Figure 5D,E and Appendix A).

## 3. Discussion

In the present study, shuterin was found to enhance the cytotoxicity of KHYG-1 cells. It exerted considerable cytotoxic effects on K562 cells by enhancing the proliferation of KHYG-1 cells and increasing the expression levels of granzyme A, granzyme B, and IFN-γ. The effect of shuterin on the cytolytic activity of KHYG-1 cells was mediated by the upregulation of the LLT1 receptor and MAPK phosphorylation and the phosphorylation of the proteins involved in the Ras/Raf pathway. Even in NK cells obtained from adult patients with HNSCC, shuterin markedly enhanced cytolytic activity through the same signaling pathway.

NK cells account for 10 to 15% of the total human lymphocytes (CD3^−^CD56^+^ cells) [31] and have wide applications in immunotherapy. However, some limitations in their use have been reported. For example, the ability for proliferation and cytotoxic activity is insufficient. Safe strategies must be devised to effectively enhance NK cell proliferation and cytolytic activity [7]. Human permanent NK cell lines are characterized by stable expansion in vitro and maintenance in vitro. KHYG-1 is a cell line that carries a p53 point mutation. Studies have shown that it is more cytotoxic to the K562 cell line than the NK-92, YT, and SNT-8 cell lines; it also has the same cytotoxic effect to other leukemia cell lines EM2, EM3, and HL60. KHYG-1 cells express NKG2D, the NKp44 receptor, and the adopter DAP12, which can be a model for studying the mechanisms of normal NK cell-mediated cytotoxicity. KHYG-1 may be a feasible immunotherapeutic agent for the treatment of cancers [20,32,33,34,35].

To enhance NK cell–based immunotherapy, we investigated the regulatory effects of shuterin on NK cells. Shuterin enhanced cell proliferation, cell cytotoxicity, protease cytotoxicity, and IFN-γ expression (Figure 1). NK cells are involved in two apoptotic pathways. They release cytotoxic granular contents (e.g., perforin, granzyme, and granulysin) into the intercellular space between NK and target cells to trigger apoptosis in the target cells. In addition, they produce cytotoxic cytokines, such as IFN-γ. IFN-γ regulates the members of the tumor necrosis factor-α cytokine family: FasL and TRAIL [36,37]. In the present study, shuterin induced intracellular signaling pathways to enhance the cytolytic activity of KHYG-1 cells.

Shuterin substantially increased the expression of granzyme B in KHYG-1 cells. Hence, we explored the mechanisms underlying the upregulation of granzyme B expression by shuterin. Subsequently, we investigated whether shuterin influenced chromatin regulation and gene expression in eukaryotic cells. Shuterin considerably increased histone acetylation; combined treatment with TSA increased this effect further (Figure 2C–F). However, this combined treatment did not enhance the expression of granzyme B in the KHYG-1 cells. Our findings are similar to previous findings on nobiletin-treated KHYG-1 cells [23]. Shuterin increases cytolytic activity by increasing granzyme B expression and is not directly associated with increased histone acetylation. The role of shuterin in increasing histone acetylation is unknown. We further evaluated the effects of shuterin on the phosphorylation of CREB, which regulates the transcription of granzyme B [11,25]. Shuterin appeared to activate granzyme B transcription by increasing the levels of phosphorylated CREB (Figure 2A,B).

NK cells induce apoptosis by releasing cytotoxic granules into the intercellular space between NK cells and their target cells. Subsequently, these granules enter the target cells by three avenues: granzymes and perforin bind to the surface of the target cells [38], mannose 6-phosphate receptors assist in the uptake of granzyme B into the target cells [39], and granzymes are absorbed into the target cells through receptor-mediated endocytosis [40]. Shuterin does not influence perforin expression or granzyme uptake. Granzyme B plays the following roles: the activation of caspase-3 and other caspases to induce apoptosis [41] and the induction of the release of proapoptotic molecules (e.g., cytochrome c) from the mitochondria by cleaving the Bid protein [42,43]. The released cytochrome c binds to apoptotic protease–activating factor-1 to form procaspase-9 and activates caspase-3 through autocatalytic activation. Granzyme B further disrupts mitochondrial membrane potential and inactivates the inhibitor of caspase-activated deoxyribonuclease through an unknown mechanism [44]. In addition, granzyme A and granulysin disrupt mitochondrial membrane potential [45,46]. We cocultured KHYG-1 and K562 cells and found that shuterin stimulated the KHYG-1 cells to release cytotoxic granules. As shown in Figure 3, our findings corroborate those in the literature that shuterin alters mitochondrial membrane potential and activates caspase-mediated apoptosis in K562 cells.

The PI3K/AKT–mammalian target of the rapamycin pathway is essential for the proliferation and activation of NK cells [47,48]. We evaluated the expression of phosphorylated AKT in KHYG-1 cells. As shown in Figure 4, shuterin inhibited the expression of phosphorylated AKT; combined treatment with inhibitors enhanced this effect on AKT phosphorylation and granzyme B. Notably, the findings varied from those obtained using the cajanine derivative LJ101019C [48]. The cell lines used in the study were different. MAPK signaling affects the cytotoxicity of immune cells (such as NK cells), exocytosis of cytotoxic granules, and expression of granzyme B [28,49,50,51]. In the present study, shuterin increased granzyme B expression in KHYG-1 cells by upregulating MAPK phosphorylation.

As shown in Figure 1D, shuterin enhanced the secretion of IFN-γ from KHYG-1 cells. We explored the signaling cascade associated with a shuterin-mediated increase in the secretion of IFN-γ from KHYG-1 cells. The LLT1 receptor reportedly induces human NK cells to secrete IFN-γ; the Ras/Raf, MEK⁄ERK, and p38 MAPK signaling pathways are essential for LLT1-stimulated IFN-γ secretion [30,52,53]. As shown in Figure 5A,B, shuterin increased IFN-γ secretion through the same pathway. HNSCC is a common type of cancer. Conventional treatment strategies for HNSCC include surgery, radiotherapy, and chemotherapy. Owing to advancements in treatment modalities, the survival rate of patients with HNSCC increased from 55 to 66% between 1996 and 2006. The poor survival rate associated with HNSCC can be attributed to laryngeal cancer and old age [54,55,56]. In the present study, shuterin markedly improved the cytotoxicity of KHYG-1 cells. We further investigated the effects of shuterin on NK cells obtained from three adult patients with HNSCC. As shown in Figure 5C,D and Appendix A, shuterin increased the release of cytotoxic granules through the phosphorylated CREB and ERK signaling pathways. In summary, shuterin increased LLT1 expression, activated the Ras/Raf signaling pathway, and substantially increased IFN-γ expression.

## 4. Materials and Methods

### 4.1. Chemicals and Reagents

Shuterin (purity ≥ 98%) was obtained from ChemFaces (Wuhan, China), and calcein AM was purchased from AAT Bioquest. The stock concentrations of shuterin and calcein AM were dissolved in dimethyl sulfoxide and stored at −20 °C. The chemical reagents used in this study, such as histone deacetylase inhibitor (trichostatin A [TSA]), granzyme B inhibitor (Z-AAD-CMK), protease inhibitor cocktail, and phosphatase inhibitor cocktail, were obtained from Sigma-Aldrich (St Louis, MO, USA). MAPK pathway inhibitors for protein kinase B (AKT; LY-294002), extracellular signal-regulated kinase (ERK)1/2 (U0126), c-Jun N-terminal kinase (JNK; SP-600125), and P38 (SB-203580) were purchased from Santa Cruz Biotechnology (Santa Cruz, CA, USA).

### 4.2. Cell Culture

The human NK leukemia cell line KHYG-1 (Japanese Collection of Research Bioresources [JCRB] accession no. JCRB0156) and the human chronic myelogenous leukemia cell line K562 (JCRB0019) were obtained from JCRB (Japan). KHYG-1 cells were cultured in Roswell Park Memorial Institute (RPMI) 1640 (Life Technologies, Grand Island, NY, USA) medium supplemented with 10% fetal bovine serum (Invitrogen, Waltham, MA, USA), 100 U/mL recombinant human interleukin (IL)-2 (Cat. #200-02; PeproTech), 100 U/mL penicillin G, and 100 µg/mL streptomycin sulfate. K562 cells were cultured in the same medium without IL-2. The cells were incubated at 37 °C under 5% CO_2_.

NK cells were isolated from the peripheral blood of adult patients with head and neck squamous cell carcinoma (HNSCC). Peripheral blood mononuclear cells were cultured using BINKIT (Biotherapy Institute of Japan, Tsukuba, Japan) according to the manufacturer’s instructions. The subculture medium comprised 10% heat-inactivated autologous plasma and 400 U/mL recombinant human IL-2. All experiments were performed on day 21 of culture.

### 4.3. Cell Proliferation Assay

The effects of shuterin on the proliferation of KHYG-1 and K562 cells were evaluated using WST-8 assays (Beyotime Technology, Shanghai, China). The cells were seeded onto 96-well plates (5 × 10^4^ cells/well), treated with various doses of shuterin, and cultured with or without IL-2 for 24 or 72 h. Finally, absorbance was measured at 450 nm to assess cell proliferation.

### 4.4. Cell Cytotoxicity Assay

Cell cytotoxicity was assessed using calcein AM. K562 cells were stained by incubating them in complete RPMI 1640 media with 10 µM calcein AM for 30 min at 37 °C under 5% CO_2_. The cells (target cells) were then seeded onto 96-well plates. KHYG-1 cells (effector cells) were treated with shuterin (10 µM) for 24 h or left untreated. Then, the KHYG-1 and K562 cells were cocultured for 2 h. Fluorescence intensity was measured at the excitation/mission wavelength of 485/530 nm. Percent lysis was calculated using the following formula: [(experimental release − spontaneous release)/(maximum release − spontaneous release)] × 100.

### 4.5. Cytokine Secretion

KHYG-1 cells were centrifuged at 190× *g* for 5 min at 4 °C. The supernatant was obtained to determine the levels of IFN-γ using LEGEND MAX Human IFN-γ ELISA Kit (BioLegend, San Diego, CA) per the manufacturer’s instructions.

### 4.6. Western Blotting

Western blotting was performed per a previously described method [57]. In brief, KHYG-1 and K562 cells were treated with shuterin for 24 h and then lysed using radioimmunoprecipitation assay buffer. The lysates were separated through sodium dodecyl sulfate polyacrylamide gel electrophoresis and transferred onto polyvinylidene fluoride membranes (Millipore, Bedford, MA, USA). The membranes were incubated with 5% skim milk for 1 h and then stained with primary antibodies overnight at 4 °C. Subsequently, peroxidase-conjugated secondary antibodies were added to the membranes and incubated for 60 min. The membranes were then subjected to ECL detection and photographed using ImageQuant LAS 4000 (GE Healthcare, Berlin, Germany). ImageJ was used for protein quantification.

### 4.7. Mitochondrial Membrane Potential Measurement

The JC-1 fluorescent dye was used to measure mitochondrial membrane potential. KHYG-1 cells were treated with various doses of shuterin for 24 h at 37 °C under 5% CO_2_. After treatment, the cells were added to the upper wells of a Transwell insert (Greiner Bio-One, Monroe, NC, USA); K562 cells (in RPMI 1640 media) were added to the lower wells of this insert and incubated according to previously described coculture methods [58]. The effector cells were cocultured with the target cells at an effector:target (E:T) ratio of 6:1 for 2 h at 37 °C under 5% CO_2_. Subsequently, the K562 cells were incubated with JC-1 solution for 15 min at 37 °C. The BD Accuri C6 flow cytometer was used for data collection, and the corresponding software was used for data analysis.

### 4.8. Statistical Analysis

Each experiment was performed at least thrice. Data are presented in terms of the mean ± standard deviation. One-way analysis of variance and Student’s *t* tests were performed. A *p* value of <0.05 indicated statistical significance. Sigma-Stat 2.0 (Jandel Scientific, San Rafael, CA, USA) was used for the data analysis.

## 5. Conclusions

Shuterin induces the production of cytotoxic granules in KHYG-1 cells through the phosphorylated CERB and MAPK signaling pathways. In addition, a shuterin-induced increase in LLT1 receptor expression and activation of the Ras-Raf signaling pathway resulted in the increased secretion of IFN-γ from KHYG-1 cells. Shuterin exhibits similar activity and efficacy, even in NK cells obtained from adult patients with HNSCC. Thus, shuterin represents a promising agent for promoting the proliferation and cytotoxicity of NK cells.

## Figures and Tables

**Figure 1 ijms-23-12816-f001:**
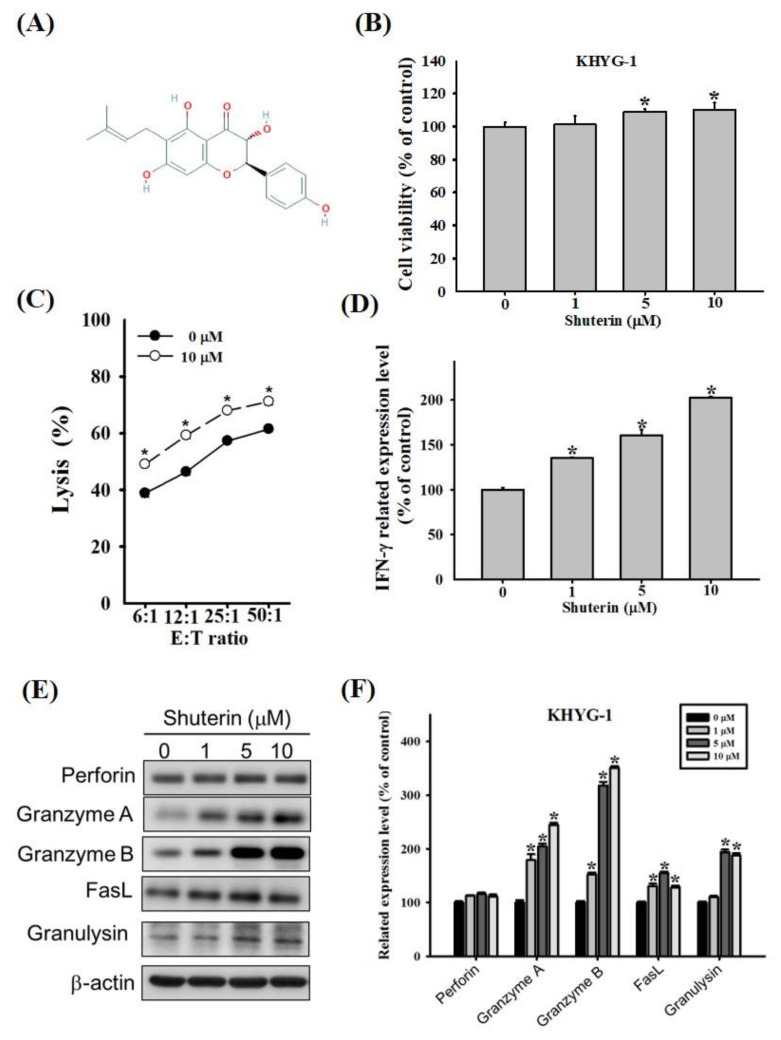
**Shuterin enhanced the cytolytic activity of KHYG-1 cells.** (**A**) Chemical structure of shuterin. (**B**) KHYG-1 cells were treated with shuterin for 72 h and then subjected to WST-8 assays for cell viability assessment. (**C**) Cytolytic effects of KHYG-1 cells treated with (or without) 10 µM shuterin for 24 h on K562 cells. Cytotoxicity was determined through the Calcein AM release assay. (**D**) Levels of IFN-γ in the culture supernatants of KHYG-1 cells treated with the indicated dose of shuterin for 24 h, then measured through enzyme-linked immunosorbent assay (ELISA). (**E**) After 24 h of shuterin treatment, the cytotoxic effectors of the KHYG-1 cells were analyzed through western blotting. (**F**) ImageJ was used for protein quantification; the levels of all proteins were normalized to that of β-actin. Data are presented as the mean ± standard deviation (n = 3) of three independent experiments. * *p* < 0.05 V.S. control.

**Figure 2 ijms-23-12816-f002:**
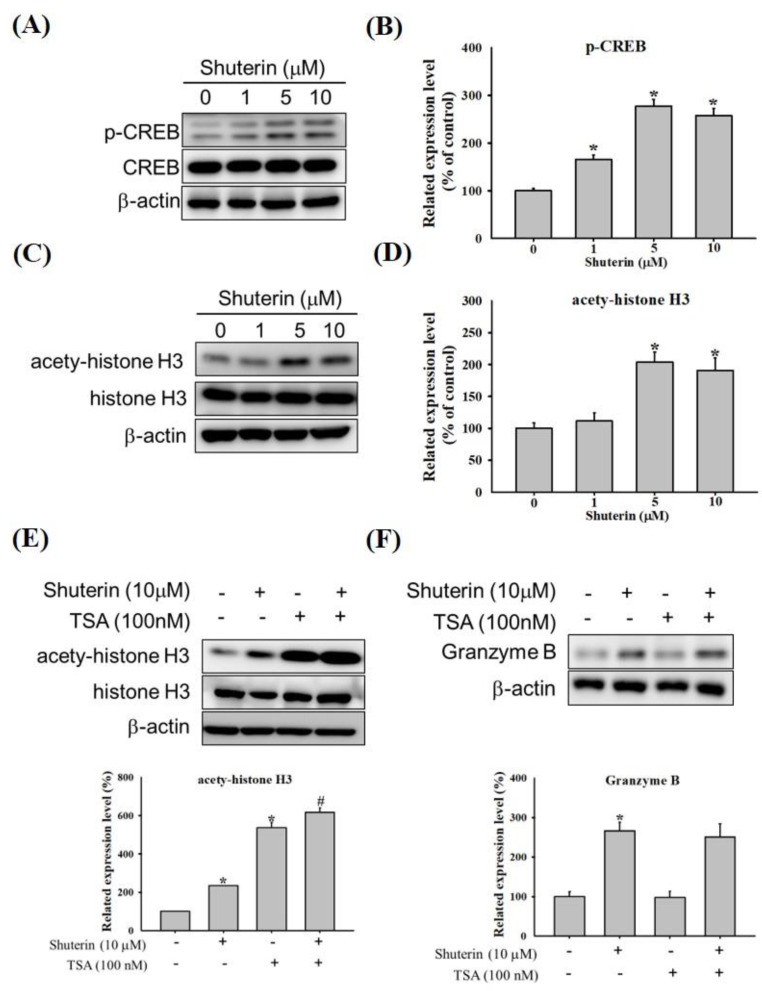
**Shuterin increased the activated CREB and histone acetylation in KHYG-1 cells.** The cells were treated with shuterin for 24 h. Protein levels of phosphorylated CREB/CREB (**A**,**B**) and acetyl-histone H3/histone H3 (**C**,**D**). KHYG-1 cells were pretreated with TSA for 1 h or left untreated; subsequently, the cells were treated with shuterin for 24 h. Protein levels of acetyl-histone H3/histone H3 (**E**) and granzyme B (**F**). Protein levels were determined using densitometry and normalized to beta-actin levels. Data are presented as the mean ± standard deviation (n = 3). * *p* < 0.05 V.S. control, # *p* < 0.05 V.S. TSA alone. CREB, cyclic adenosine monophosphate response element–binding protein; TSA, trichostatin A.

**Figure 3 ijms-23-12816-f003:**
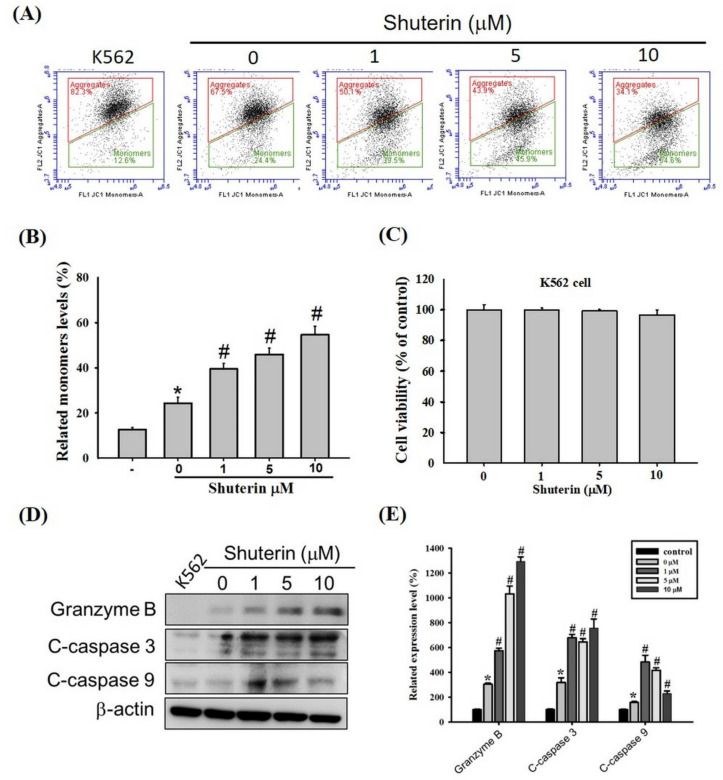
Shuterin-induced apoptosis in cocultured K562 cells. (**A**) The mitochondrial membrane potential of K562 cells was measured. (**B**) Quantitative data were analyzed using BD Accuri C6 software. (**C**) K562 cells were treated with shuterin (0–10 μM) for 24 h. Cell viability was assessed using WST-8 assays. (**D**) K562 cells were cocultured with KHYG-1 cells at an effector:target ratio of 6:1 and then treated with shuterin (0–10 µM) for 24 h. (**E**) Compared with control cells, statistically significant results were obtained in treated cells. * *p* < 0.05 V.S. control, # *p* < 0.05 V.S. cotreatment control.

**Figure 4 ijms-23-12816-f004:**
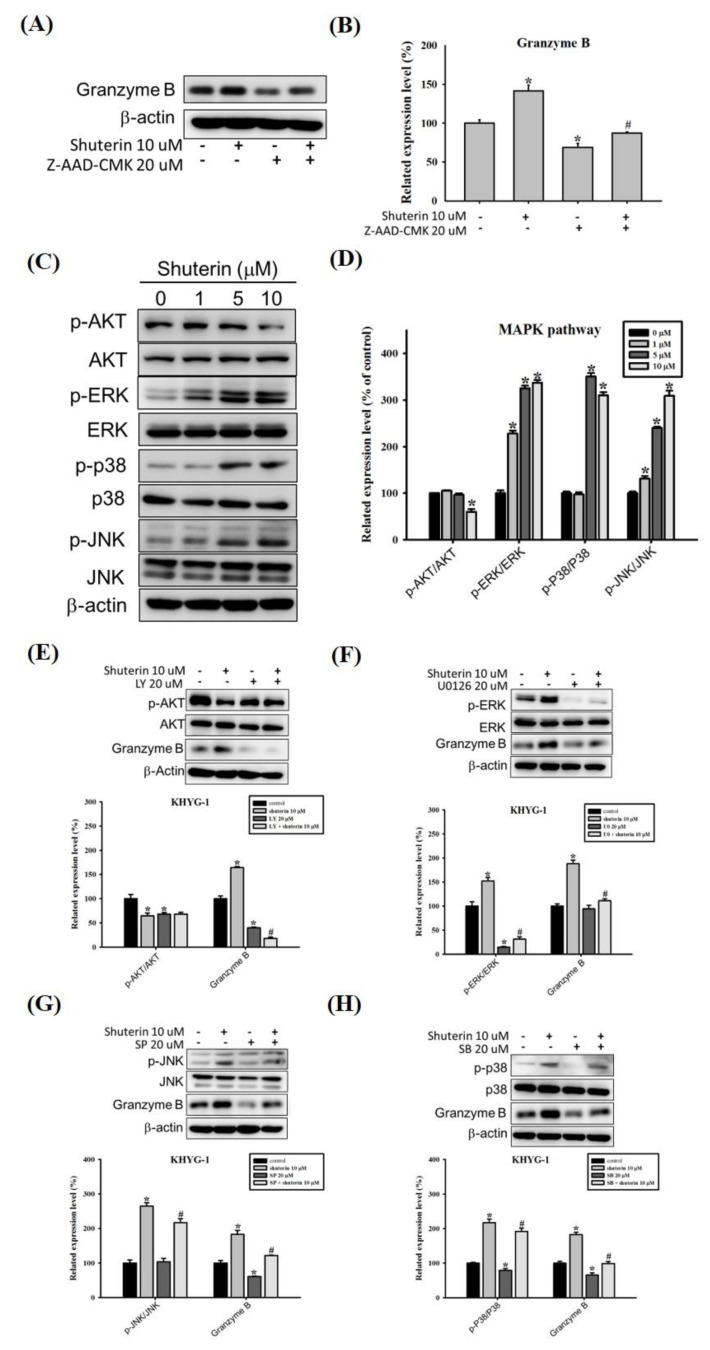
Mechanisms underlying the shuterin-induced increase in granzyme B expression through the MAPK pathway. Cells were pretreated with various inhibitors (Z-AAD-CMK/granzyme b, LY-294002/AKT, U0126/ERK, SP-600125/JNK, and SB-203580/P38) for 1 h or left untreated; subsequently, these cells were treated with shuterin for 24 h. Protein levels were quantified through Western blotting and normalized to beta-actin levels using a densitometer. Granzyme B and granzyme b inhibitors (**A**,**B**); PI3K/AKT and MAPK pathway (**C**,**D**); p-AKT/AKT (**E**); p-ERK/ERK (**F**); p-JNK/JNK (**G**) and p-p38/p38 (**H**) * *p* < 0.05 V.S. control, # *p* < 0.05 V.S. inhibitor-treated cells.

**Figure 5 ijms-23-12816-f005:**
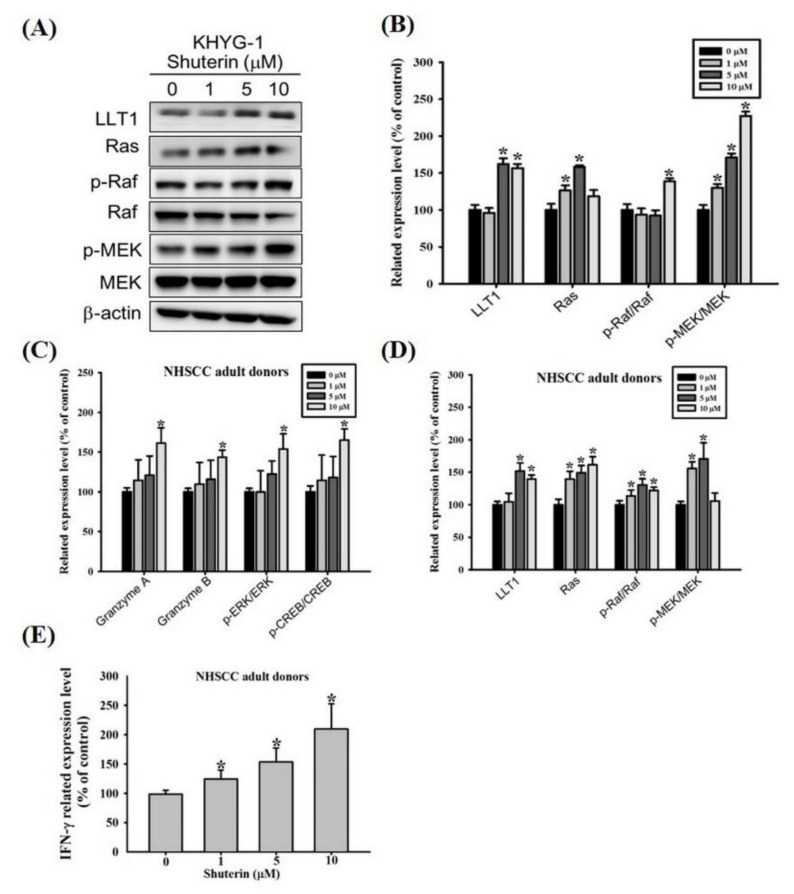
Shuterin induced the Ras/Raf pathway and increased IFN-γ expression in KHYG-1 cells and NK cells obtained from adult patients with head and neck squamous cell carcinoma (HNSCC). Effects of shuterin treatment (24 h) on the proteins involved in the Ras/Raf pathway in (**A**,**B**) KHYG-1 cells and (**C**,**D**) NK cells obtained from adult patients with HNSCC. ImageJ was used for protein quantification; the levels of all proteins were normalized to those of beta-actin. Data are presented as the mean ± standard deviation (n = 3). * *p* < 0.05 V.S. control. (**E**) Levels of interferon (IFN)-γ in shuterin-treated (24 h) NK cells that were obtained from adult patients with HNSCC (measured through ELISA).

## Data Availability

This study did not report any data.

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
