# Peer review of "Shuterin Enhances the Cytotoxicity of the Natural Killer Leukemia Cell Line KHYG-1 by Increasing the Expression Levels of Granzyme B and IFN-γ through the MAPK and Ras/Raf Signaling Pathways"

_ijms, 2022, doi:10.3390/ijms232112816_

Round 1

Reviewer 1 Report

The paper is  an impressive and convincing demonstration of the immunosuppressive potential of a natural phytochemical on NK cells .  Although the clearly presented results, the significance of the study is questionable for following reasons:

1. Shuterin is a quite unknown phytochemical. The authors should provide a detailed description of the molecule, e.g. it belongs to the group of phytoalexins. 

2. Why did the authors choose this phytochemical for their experiments? 

3. What do we know about shuterin as a drug candidate? Is there any information about the application possibilities of the substance and bioavailability? Experiences in animals? How were the concentrations used in the experiment determined?

4. Please give examples for NK cells according your your statement about cancer immunotherapy: "Immune cells, such as natural killer (NK) cells (e.g., γδ and αβ T cells), cytotoxic T lymphocytes (CTLs), and dendritic cells, have various clinical applications."

5. Please provide references for your statements: "NK cells....have wide applications in immunotherapy. However, some limitations in their use  have been reported." 

Reviewer 2 Report

The study by Li et al. evaluates the effects of a phytochemical shuterin on NK cell functions. The study is compact and shows some novel effects of shuterin. It is not always clear how the authors came to perform some of the experiments. I have a few concerns.

1. Fig.1 Shuterin increases proliferation of KHYG-1 cells which are per se a cancerous cell line. The authors should comment on that while discussing its potential application in the context of immunotherapy

2. Line 104. It is completely unclear why the authors at this point of the paper want to study general histone acetylation. TSA usage is not explained

3. Line 134: Why do the authors focus on MAPK pathway directly? If other pathways regulating GzmB expression have been tested it would add a lot of value to also show negative data on other signaling pathways

4, Figure 5: It is necessary to show not only the quantifications but the WB from the primary NK cells. Why did the authors not use primary NK cells from healthy donors? This would have been an important control. The authors should also explain more in detail why the NK cells from such patients were used.

5. Line 163: This sentence is not reflected in the data shown. The IFNg is associated with higher Raf but its expression is not necessarily increased via this pathway.

6. How is the standard deviation achieved in quantification of WB? Where these multiple experiments? Or just multiple exposures and measurements? This should be stated.

7. The effects on Gzm, perforin and expression of many other proteins is shown but no transcriptome data is provided. Are these changes transcriptional? This is an important experiment

Round 2

Reviewer 2 Report

The authors addressed my comments properly. However the newly added text (red) requires extensive correction of English.